# Study of Rail Squat Characteristics through Analysis of Train Axle Box Acceleration Frequency

## Hojin Cho * and Jaehak Park

Global Solution Group, Daedeok-gu, Daejeon 302 105, Korea; jaehak@igsg.co.kr
* Correspondence: hojin@igsg.co.kr; Tel.: +88-70-4922-5116

**Abstract:** In this study, a method for detecting the railway surface defects called "squats" using the ABA (Axle Box Acceleration) measurement of trains was proposed. ABA prototype design, implementation, and field tests were conducted to derive and verify the results. The field test was performed using a proven precision measurement system, and the measured data were signal-processed using a Matlab program. The algorithm used to determine the position of the squats was developed based on wavelet spectrum analysis. This study was verified for a section of a domestic general line and, following field verification for the section, squats was detected with a hit rate of about 88.2%. The main locations where the squats occurred were the rail welds and the joint section, and it was confirmed that in some sections, unsupported sleepers occurred at the locations where the squats occurred.

**Keywords:** Axle Box Acceleration (ABA); rail transportation maintenance; railway monitoring; surface defects on railway rails





## 1. Introduction

Early detection of abnormal railway behavior and rapid performance of maintenance work can prevent traffic blocks and minimize long-term costs of railway infrastructure. The present study was conducted by focusing on the detection of squats, which are small defects that start on the rail surface [1–3]. Squats begin as a small groove, corrugation, or weld portion [4,5]. When squats are detected in their early stage and the deterioration is mild, maintenance may be easily performed by polishing a thin layer of the rail surface. Since severe squats may require to replacement of the rail, early detection of squats may significantly reduce rail maintenance cost.

The total railway length in Korea (excluding urban railways) is about 2750 km, including 3400 bridges, 840 tunnels, and 1241 stations. This system transports over 420,000 passengers via 2000 trains each day. The cost of replacing a rail is high, and train delay or derailment due to defects of rails can give the public a negative impression of the railway system. Therefore, it is critical to prevent disruptions of system operation.

Keeping rail in good condition requires systematic and periodic testing. A computer-based tool is used for such tests, and rail managers implement systematic maintenance to minimize the total cost and secure the long-term quality of the rail [6]. Various methods can be used to monitor rail track conditions [7–11].

Visual inspection and ultrasonic and eddy current measurement methods are used in Korea and many other countries to detect short-wavelength defects of railways [12,13]. However, these types of tests are effective only in cases in which the rail performance is decreased due to severe damage; they are not optimal methods for maintenance. In addition, visual inspection requires a large amount of labor, and different results may be derived by different workers [14]. Therefore, it is necessary to develop a method to automatically detect irregular defects such as squats, and a monitoring system to efficiently evaluate the conditions of tracks, including rails.

In the Netherlands, axle box acceleration (ABA) has been used since the mid-1980s to detect defects such as rail squats and poor welding [15]. Compared to other methods, ABA costs less and is easy to maintain. Recently, accelerometer-based methods have been proposed to monitor rail conditions [16–19]. Lee et al. [20] suggested a mixed filtering (Kalman filter and band-pass filter) method for estimating the irregularity of rails by using accelerometers mounted on wheel axles and bogies. Shafiullah et al. [21] presented a communication protocol between accelerometers installed in a train system to monitor the typical dynamic behavior of a train. The ABA method has been used to detect rail defects, including corrugations and welds of rails as well as of insulation joints (seams) [22–24]. However, the ABA-based measurement method is not capable of detecting all types of squats. Extensive detection is impossible because squats are usually generated randomly and only one squat is generated at a position. In addition, the ABA signals (especially light rail squats) may not be easily detected and analyzed without appropriate equipment and signal processing.

This paper presents squat detection results obtained from the system applied in the present study, which was realized and verified through on-site tests on general railways in Korea. The system developed in the present study can be mounted on trains in operation to perform continuous monitoring of the entire railway infrastructure at a relatively low cost through continuous updates. In addition, the developed system is suitable for the task of addressing dramatic performance decreases, after which appropriate maintenance may not be applied due to the long cycle of general tests. The system developed in the present study may be applied to various commercial trains to significantly improve the safety of railway operation and reduce the life-cycle cost of the entire infrastructure.

## 2. Description of System
*Squats*

Squats, which generally start from a surface, are rolling contact fatigue (RCF) defects of rails. Squats are caused by the repeated loading on the narrow contact surface in the middle of the rail heads by the wheels on the straight sections of the rails. Figure 1 shows the three types of squats, which are light, moderate, and severe squats [25].

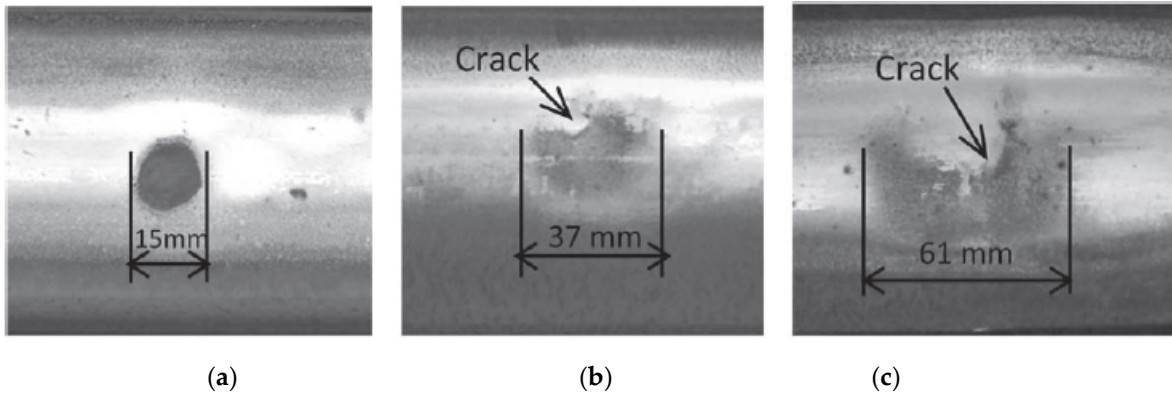

**(a)**　　　　　　　　　　　　　**(b)**　　　　　　　　　　　　　**(c)**

**Figure 1.** (**a**) Light squat. (**b**) Moderate squat. (**c**) Severe squat [25].

Generally, squats are not so deep as to touch the wheel (deeper than the depth of compression of the rail by the wheel load (0.05 mm)). Some squats cause cracks and become rusty.

Squats are expanded by the dynamic contact between the wheel and the rail. The squat length is generally 20 to 40 mm, but it can increase to 60 mm depending on the rail strength [1]. Squats are reflected to the wheel vibration characteristics through dynamic wheel–rail interaction, and thus can be detected by an accelerometer.

Not all the defects generated on the rail surface develop into squats. A rail surface defect with a size greater than a threshold value may develop into a squat. The threshold

defect size is 6 to 8 mm in both the longitudinal and transverse directions when the train traction and braking forces are at maximum. A defect smaller than the threshold size is insignificant, and the defect may be worn out and disappear.

Realization of Measurement System and Data Verification

As shown in Figure 2, accelerometers installed on the EM-140K (detection speed 80 km/h) were used to measure the train axis acceleration. The EM-140 is a vehicle that inspects Korea's track infrastructure. Three accelerometers were mounted on the axis of the bogie, two in the vertical direction and one in the transverse direction. The DAQ measurement equipment (NI), which is reliable for frequencies higher than 1 kHz, was used in the system to measure both the vertical and longitudinal signals. The vertical signals are effective for detecting light squats because of the high signal-to-noise ratio. The sampling frequency of the accelerometers was 2 kHz. Figure 2 shows the DAQ system that was employed to measure the axis acceleration of the EM-140K. The system measures the coordinates through the GPS receiver, and the train velocity can be calculated based the coordinates. The GNSS system used in this study had an accuracy of 0.01 m + 1 ppm CEP as a high-precision GNSS RTK receiver. The maximum speed of the test vehicle was about 140 km/h and vehicle speed over the study period was about 80 km/h. Speed measurement was determined as the correct value because it produced the coordinate measured by the tachometer and the GNSS receiver attached to the train. In addition, the accuracy of the position measurement by the field inspection is less than $\pm 5$ m.

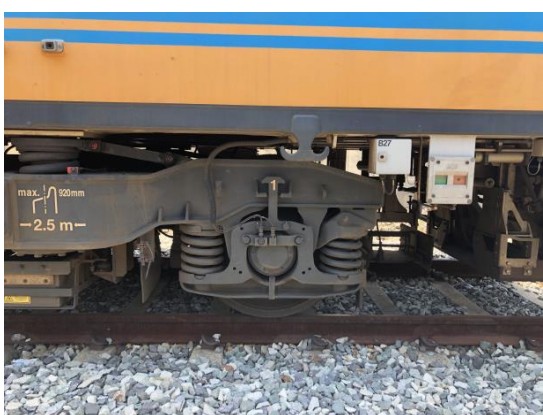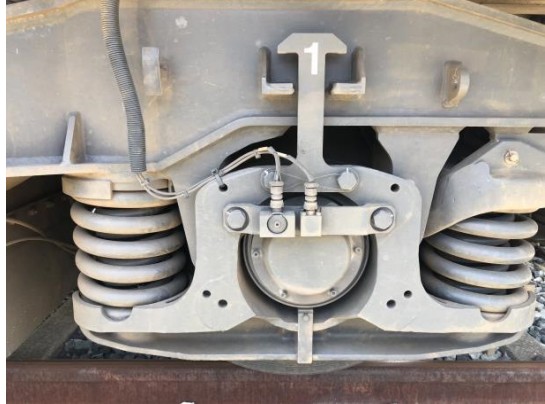

**Figure 2.** EM-140K accelerometer.

A visual inspection was performed to acquire the details of the rail over a short interval. The valid data acquired by visual inspection included GPS coordinates of defects, images, welds, and joints.

## 3. Wavelet Analysis

### 3.1. Frequency Analysis Using Wavelet Analysis

The axis acceleration measurement in the time domain is not sufficient to detect tiny defects. Various analytical methods may be used to analyze the frequency characteristics of a signal, but not all methods provide good information. The major drawback of the short-time Fourier transform (STFT) often applied to frequency characteristics analysis is that the window size must be selected. When axis acceleration is analyzed using STFT, the time and frequency resolutions are inversely proportional if the window size is changed. For example, when the window is short, the frequency resolution decreases while the time resolution increases. The detection of different squats and rail element evaluation over a wider range require different window sizes for each defect type. Therefore, STFT is not considered a suitable method.

Therefore, we applied wavelet transform by analyzing the acceleration data in the present study and did not apply STFT analysis techniques to find the squats. This technique was used to assess the condition of the track; many previous papers have applied the wavelet transform. Representative examples include Professor Zili Li's team at Delft University of Technology in the Netherlands and Hitoshi in Japan. The present study was conducted using wavelet analysis, which has the advantage that the time–frequency expression is not dependent on the window size. Continuous wavelet transform (CWT) is a time–frequency analysis method in which an observed function is multiplied by a shifted and scaled wavelet function group. There are many cases and papers where wavelet transform has been applied as a technique to evaluate the state of an orbit.

CWT is defined as in the equation below [26]:

$$W_n(S) = \sum_{n'=0}^{N-1} x_n \psi^* \left( \frac{(n'-n)\delta_t}{s} \right) \tag{1}$$

In Equation (1), $x_n$ is a time series in which the time step is $\delta_t$, and $n$ is a time index wherein $n' = 0, \ldots, N-1$ is a time shift operator. $\psi$ is a mother wavelet with locally limited functions. $\psi^*((n'-n)\delta_t/s)$ is a wavelet derived from a mother wavelet by translation and scaling. The symbol * denotes a conjugate complex number, and $s$ is a wavelet scale ($s > 0$). $W_n(s)$ is the wavelet coefficient.

The wavelet scale is related to the Fourier period (or inverse frequency). According to Parseval's energy conservation theorem, the energy of a wavelet transform is equal to the energy of the original signal in the time domain. The physical meaning of the CWT may be explained by the correlation in a delayed n between the original signal and the scaled wavelet. Wavelet transformation may be considered as a linear filter involving a parallel filter set [27]. The key advantage of CWT over STFT is the high time and frequency resolutions [18]. Therefore, CWT is suitable for investigation of local changes of frequency components, including the detection of structural damage and defects and identification of cracks and other abnormal phenomena [27–29]. In the present study, the Morlet function was employed as the mother wavelet.

The Morlet function is defined as in Equation (2):

$$\psi_0(\eta) = \pi^{-1/4} e^{iw_0\eta} e^{-\eta^2/2} \tag{2}$$

where $\omega_0$ is a dimensionless frequency. The power spectrum of a wavelet transform is the square of the wavelet coefficient, which is expressed as Equation (3):

$$\left| W_n^2(s) \right| \tag{3}$$

The wavelet power spectrum (WPS) is expressed in a scalogram. The vertical slice on the right side of the wavelet plot is the scale of a general spectrum. Figure 3 shows the scalogram of the acceleration signals with respect to the squats. The red color represents a high signal energy level due to impact at a squat. In the present study, the scalogram was used to define the time–frequency relationship of the squats and the axis acceleration signals.

### 3.2. Noise Filtering

The noise filtering of measurement data was performed in two stages.

(1) Axis acceleration filtering: The data were filtered using the signal processing toolbox of the MATLAB software program. A low-pass filter with a cutoff frequency of 2000 Hz was applied to separate the response data of the squats.
(2) Filtering of noise due to wheel damage: Additional signal processing was performed when the wheel conditions were not good. Wheel damage is detected more easily than squats, because it causes a cyclic impact between the wheel and the rail at a wavelength corresponding to the wheel circumference. The signal is repeatedly

detected by the accelerometer nearest to the damaged wheel. The problem of repeated acceleration signals was addressed by removing the repeating pattern.

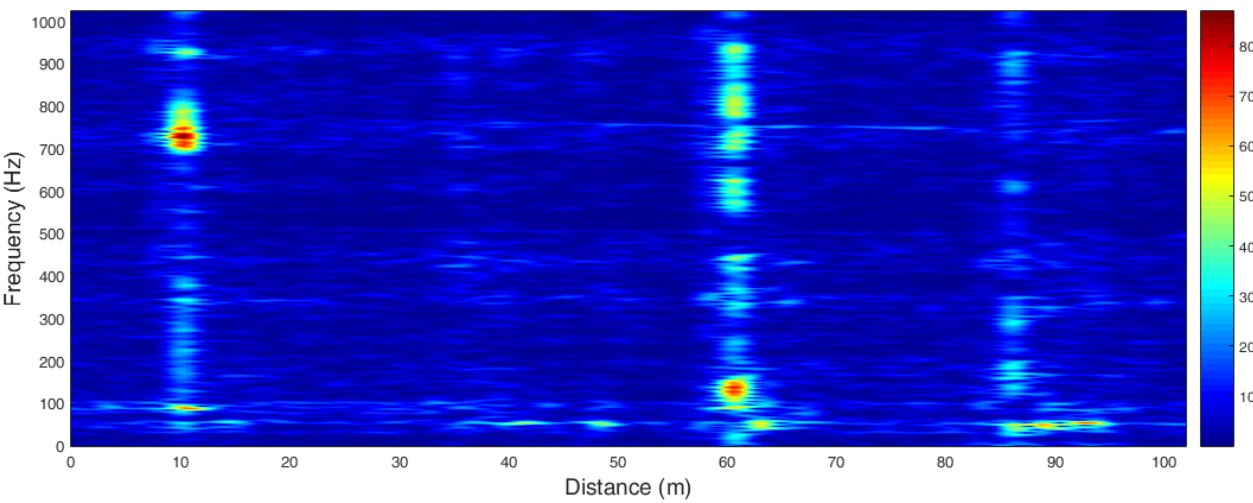

**Figure 3.** Scalogram of ABA signal.

## 4. Verification of Squat Detection

### 4.1. Data Acquisition

The present study was conducted by performing data collection on a 3 km straight railway interval subjected to mixed traffic with a track speed of approximately 80 km/h, measurement of data, preprocessing for noise filtering, and squat detection and evaluation. Figure 4 shows the flowchart of the squat detection algorithm.

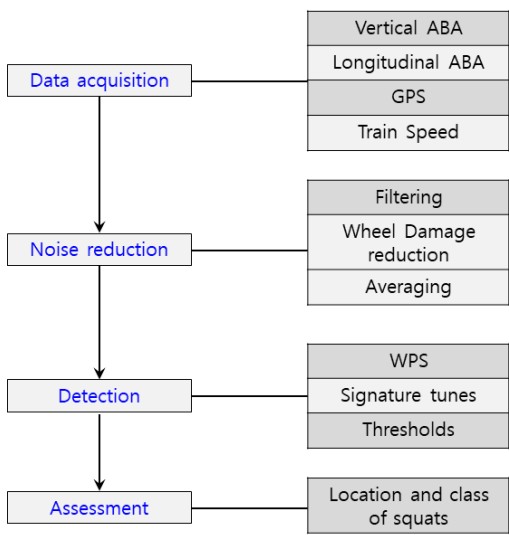

**Figure 4.** Detection procedure.

The test section used in this study was a Korean general rail section (passenger/cargo mixed section). The speed during the test was about 80 km/h and the vehicle was driven at a constant speed. In order to focus on the contents of the squat, data from a section without a special track system such as a turnaround system were used.

Figure 5 shows the acceleration data acquired from a general railway in Korea, presenting data acquired for about 5 s from four representative sections. In all sections, the magnitude of the acceleration was about $\pm 5$ m/s$^2$ for small vibrations and about $\pm 10$ to 15 m/s$^2$ for large vibrations.

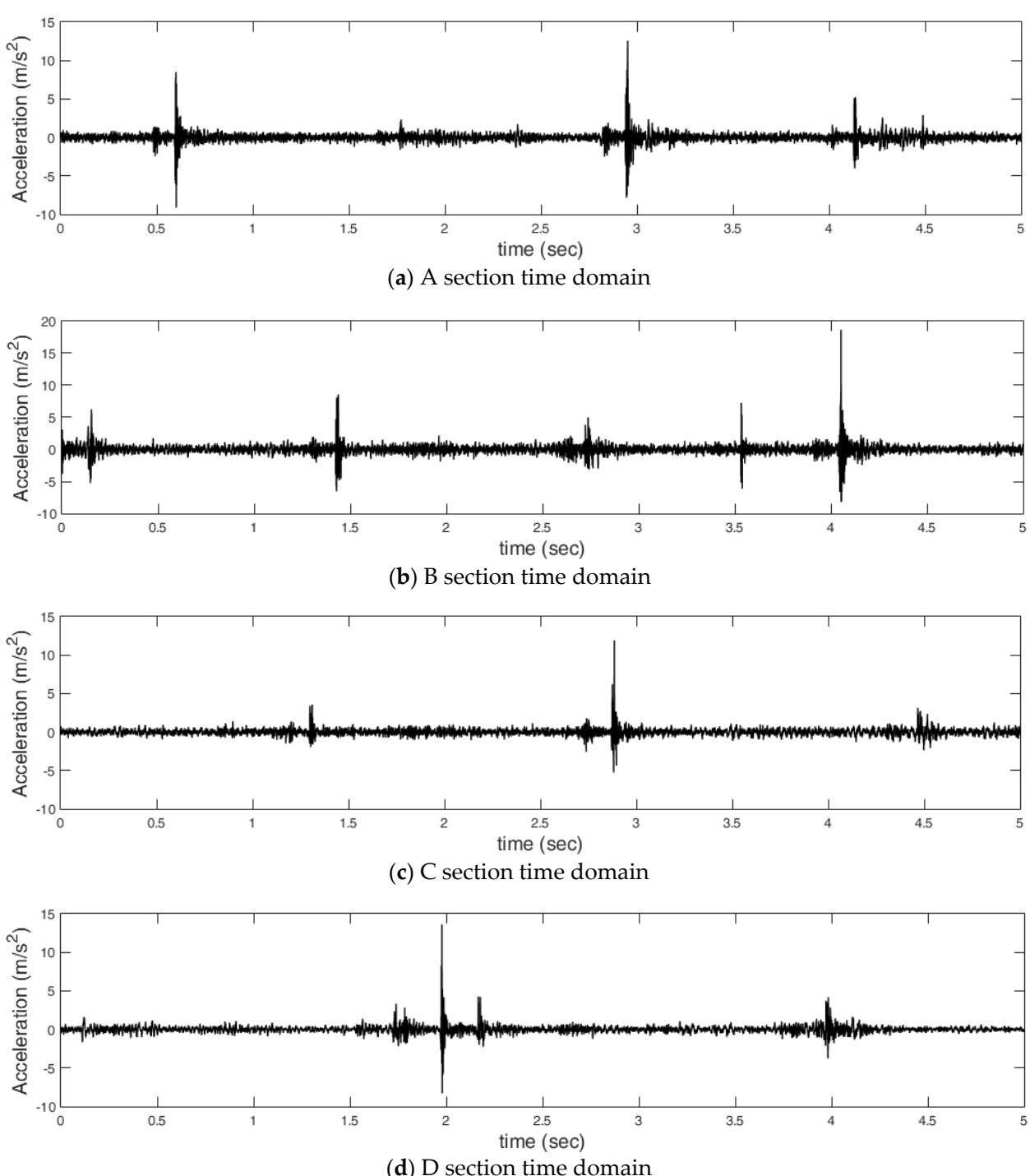

(**a**) A section time domain

(**b**) B section time domain

(**c**) C section time domain

(**d**) D section time domain

**Figure 5.** Data obtained from measurement system.

### 4.2. Frequency Characteristics of Squats

Wavelet analysis was performed using the vibrational acceleration shown in Figure 5. In the wavelet analysis, the velocity was integrated to calculate the distance and the time axis (the *x* axis) was converted to the distance axis, as shown in Figure 6. At the squats, the maximum frequency response was 1.0 kHz in both the vertical and transverse directions. The frequency band of the squats was dependent on the squat size and position, but a high WPS was usually found in bands of 100 to 200 Hz and 700 to 800 Hz.

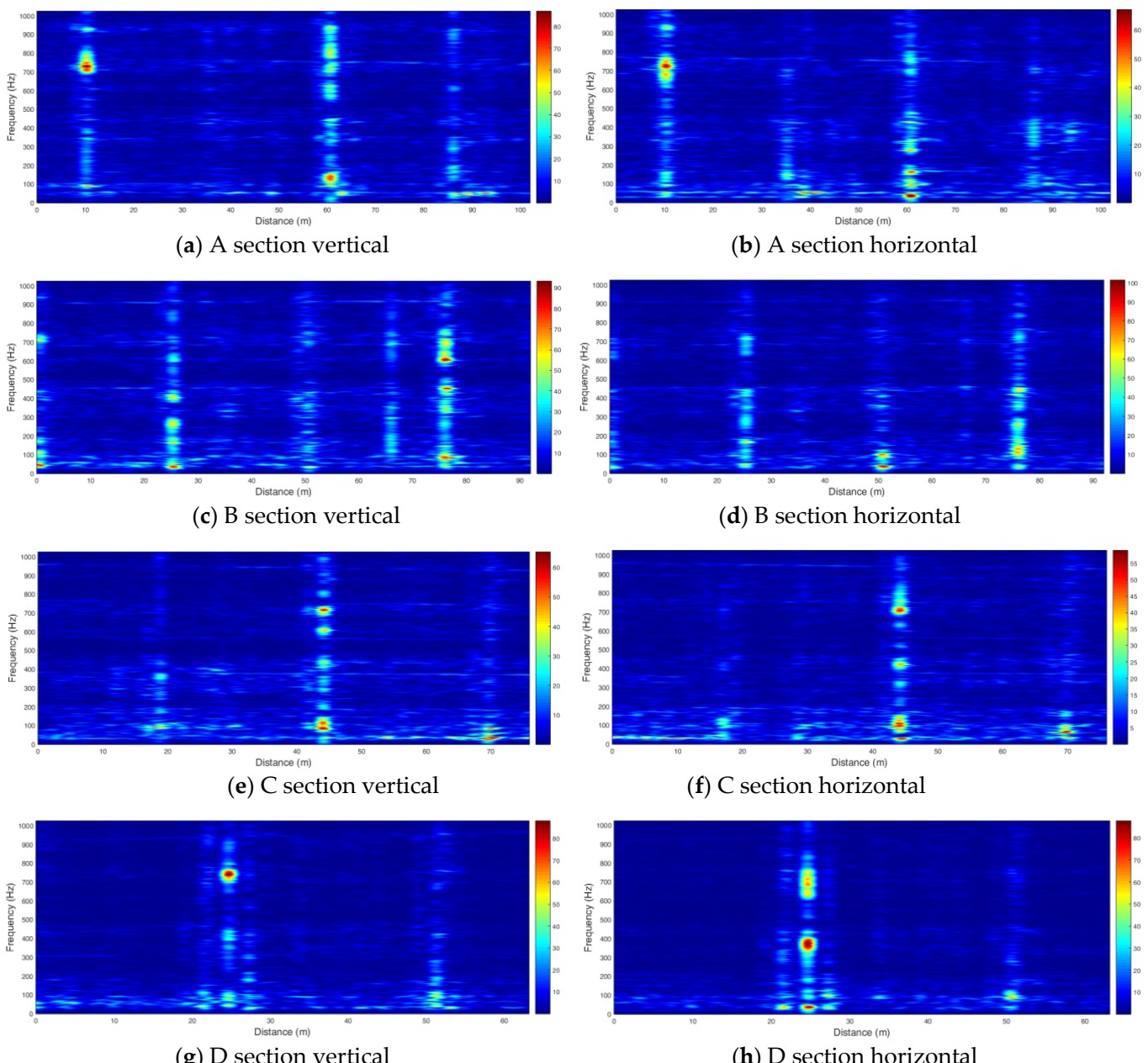

**Figure 6.** Vertical and horizontal squats.

On-site verification was performed in sections where the power spectral density (PSD) was high. The on-site verification showed that the positions where the WPS was high in the band of 700 to 800 Hz were joints, as shown in Figure 7. The positions where the WPS was high in the band of 100 to 200 Hz were squats generated at the rail welds, as shown in Figure 8. The positions where the WPS was high in relatively low frequency bands of 100 Hz or lower were found to be positions of hanging sleepers, as shown in Figure 9. The frequency band at the positions of loose sleepers was dependent on the range of the loose sleepers. Therefore, additional study may need to be conducted in this regard.

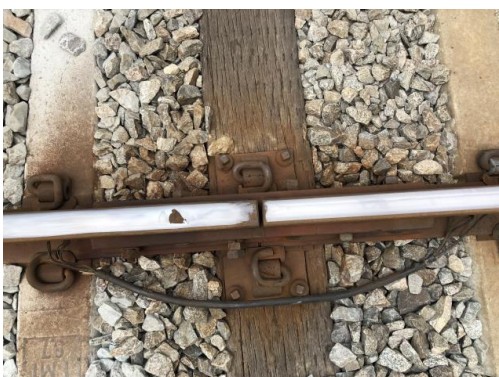

**Figure 7.** Rail joint section.

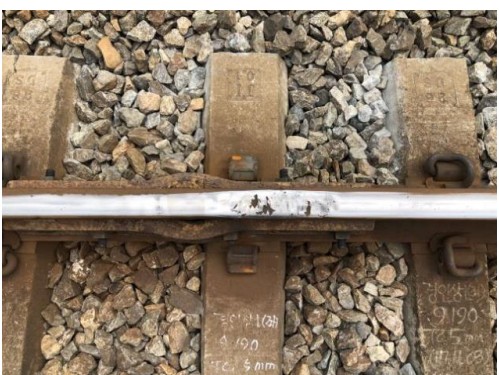

**Figure 8.** Weld section.

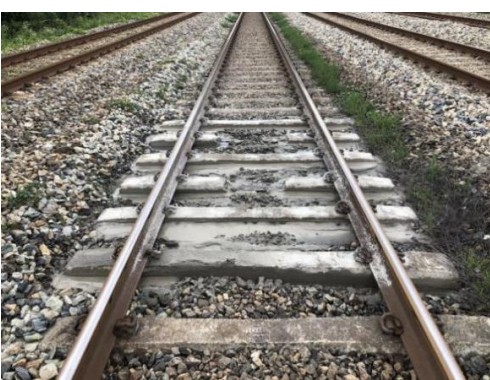

**Figure 9.** Hanging sleepers.

*4.3. Detection of Squat Defects*

The results of the study conducted on a general railway sections showed that the outstanding frequency range of the axis acceleration due to the squats was 100 to 200 Hz and 700 to 800 Hz. Therefore, WPS was high in the frequency ranges of 100 to 200 Hz and 700 to 800 Hz. These results were applied to the squat detection program developed in the present study to predict the positions of squats. The threshold of the WPS was set to 20, and the positions showing WPS values higher than the threshold were inspected visually. A total of 34 positions of squat generation were confirmed by visual inspection, while 38 positions were predicted as squat positions by the program developed in this study. Figure 10 shows the WPS in the measurement interval and the defect positions. Table 1 shows the actual number of squats and the prediction accuracy of the developed program. With reference to the 34 defect positions, the prediction accuracy was 88.2% and the false alarm rate was 10.5%.

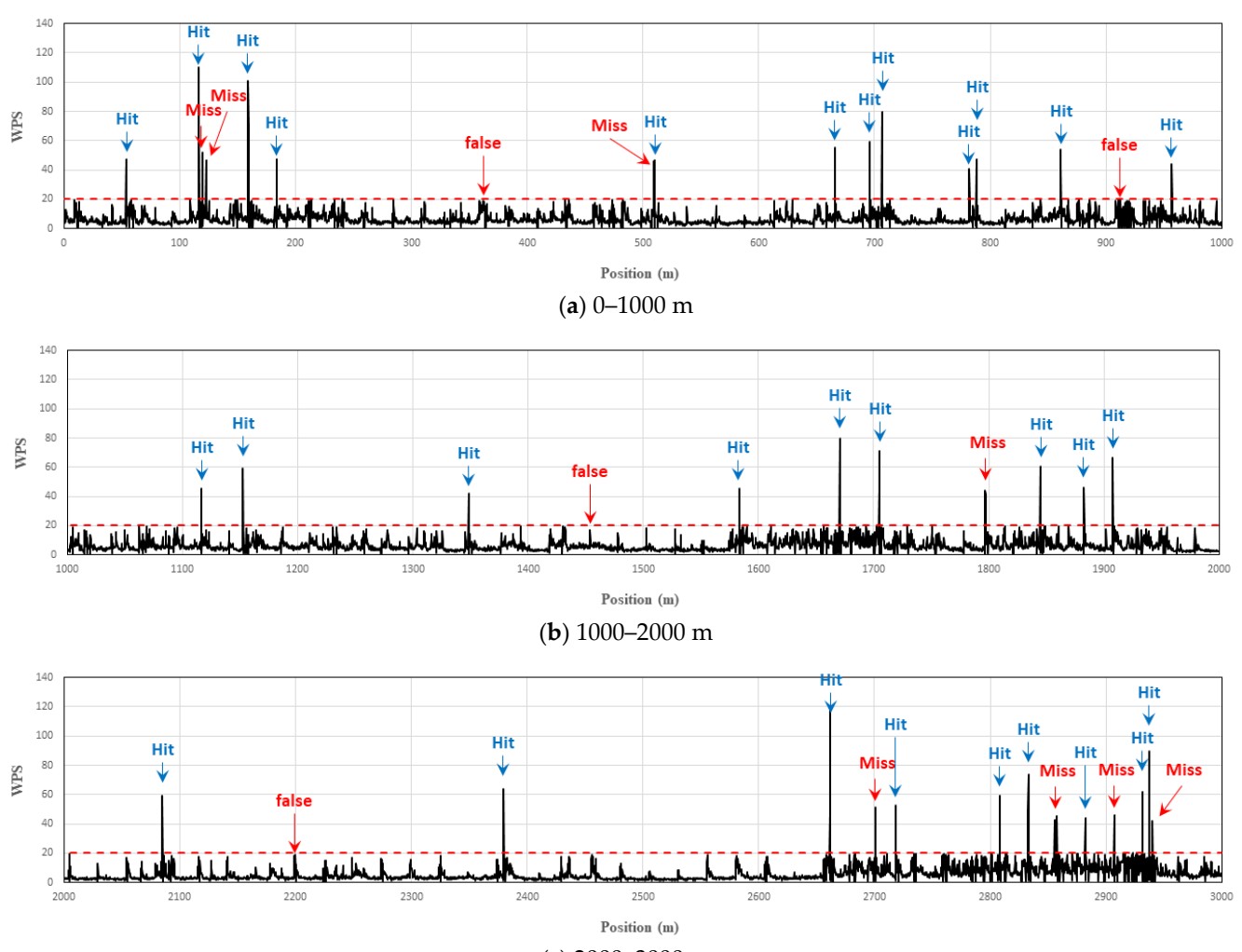

**Figure 10.** WPS values and detected positions.

**Table 1.** Design weight of tilting train.

| Location (m) | No. of Defects | No. of Predictions | No. of Detected Defects | No. of False Alarms | HR (%) | FA (%) |
|---|---|---|---|---|---|---|
| 0–1000 | 14 | 15 | 12 | 2 | 85.7 | 13.3 |
| 1000–2000 | 10 | 10 | 9 | 1 | 90.0 | 10.0 |
| 2000–3000 | 10 | 13 | 9 | 1 | 90.0 | 7.7 |
| Total | 34 | 38 | 30 | 4 | 88.2 | 10.5 |

## 5. Results

This article proposes a measurement system for detecting squats generated on railways. The squat detection program is currently in development. The results of the program were verified by conducting a visual inspection. The proposed detection program is based on the WPS of the measurement frequency. The frequency bands associated with the squats were identified using the wavelet spectrum. The frequency bands of the squats were 100 to 200 Hz and 700 to 800 Hz depending on the characteristics of the rails with the squats (joints, welds, hanging sleepers, etc.).

The threshold for detecting squats on rails was established empirically. The accuracy of squat detection by the program was 88.2% and the false alarm rate was 10.5%.

The application of the proposed measurement system with acceleration data analysis may help with the performance of efficient rail maintenance and appropriate preventive measures. The effective maintenance method suggested in this paper may significantly reduce the life-cycle cost of rails that are affected by squats. Further studies will be conducted to improve the algorithm and develop a system that can be applied to various trains to monitor rails.

**Author Contributions:** Conceptualization, H.C.; data curation, H.C.; formal analysis, H.C.; methodology, H.C. and J.P.; software, H.C.; supervision, H.C.; writing—original draft, H.C. and J.P.; writing—review & editing, H.C. and J.P. Both authors have read and agreed to the published version of the manuscript.

**Funding:** This work is supported by the Korea Agency for Infrastructure Technology Advancement(KAIA) grant (21CTAP-C152128-03).

**Institutional Review Board Statement:** Not applicable.

**Conflicts of Interest:** The authors declare no conflict of interest.

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
