# Peer review of "Study of Rail Squat Characteristics through Analysis of Train Axle Box Acceleration Frequency"

_applsci, doi:10.3390/app11157022_

Round 1
Reviewer 1 Report
The article is well written and understandable. To my opinion it can be published as-is (after the correction of the typo in line 127).
I have minor suggestions that might further improve the article (it is up to the authors to include my suggestions or not):
Line86:
optional additional sentence, e.g:
Rail track maintenance strategies are strongly different between rail infrastructure management companies in the world, many still rely on periodic visual inspections, some have advanced towards sensor driven condition monitoring approaches.
Lines 83-87
”A rail surface 83 defect having a size greater than a threshold value may develop into a squat. The thresh- old defect size is 6 to 8 mm in both the longitudinal and transverse directions when the train traction and braking forces are at maximum. A defect smaller than the threshold size is insignificant, because the defect may be worn out and disappear.”
This is not exactly true - in case of head-checks the defect sizes are lower but may still lead to catastrophic failure of the rail. I suggest to link this "threshold" to the squats defect type only.
line 97-98
“The sampling frequency 97 of the accelerometers was 2 kHz”
To give the reader a better impression on the amount of data used for the verification on the 3km long section, an indication of the velocity of train during the measurement would help (see also comment on the 3km section below)
line101
“A visual inspection was performed to acquire the details of the rail in a short interval.”
“a short interval” is not very precise – do you mean a time interval (which one? weeks, months?) or did you mean the 3km track section?”
line127
typo? Wn(s) – n should be subscript, S should be capital or small if the capital letter in the formula is wrong)
lines 165-166
“The present study was conducted by performing data collection on a 3 km general railway interval in Korea”
A bit more details would help: the track section is mainly straight or curved, subjected to what kind of traffic? mainly passenger trains, heavy haul line or mixed traffic?
more precise would be e.g.: a 3km straight railway interval subjected to mixed traffic with a track speed of approx. xxx km/h.
Final remarks:
An outlook on the next steps towards automated squats detection would round up the paper.
Without the measurement actual data a judgement of the "scientific soundness" and reproducibility of the results is hardly possible. Making the data openly accessible would be highly welcome, on the other hand it is understandable that rail infrastructure managers do not want to publish data that indicates defects in ther infrastructure in detail.
Author Response
Although this thesis is lacking in many ways, thank you very much for reviewing it. We will do our best to do better research.
Line 86
Thank you. We will do our best to refer to the points you pointed out for further research.
Lines 83-87
Thank you. We will do our best to refer to the points you pointed out for further research.
line 97-98
This content was added on line 92 of the paper.
line101
"a short interval" means that the track condition was checked in more detail.
line127
This content was added on line 128 of the paper.
lines 165-166
This content was added on line 167 of the paper.
Final remarks:
Rail infrastructure managers in Korea(KORAIL) do not want to disclose defect data on rails. Please understand.

Reviewer 2 Report
Chapter 2.1.1.
EM-140K is the name of an accelerometer or a railway vehicle? What GNSS system was used? What was the accuracy of the speed measurement? What was the accuracy of the position measurement? What were the measurement assumptions? Information on the speed of the test vehicle is missing.
Chapter 3.1.
It is stated that the STFT is not suitable for squat detection, but there is no evidence of this. There is no comparison of the results of the presented methods or references to the literature.
The article does not indicate the advantage of the presented system over others currently used.
What is the duration of the tests, what speeds were during the tests? What is the influence of braking or acceleration, geometrical system, turnouts, frogs etc...
It is necessary to develop the research part, description of the rail route, track condition, measuring car.
It is also necessary to develop the comparative part, showing the advantage of the measurement method and signal processing over other methods.
Author Response
Chapter 2.1.1.The EM-140 is a vehicle that inspects Korea's track infrastructure.
The GNSS system used in this study is a high-precision GNSS RTK industrial receiver with an accuracy of 0.01m + 1ppm CEP.
The maximum speed of the test vehicle is about 140 km/h, and the vehicle speed in this study section is about 80 km/h.
Since the speed measurement was obtained with the coordinates measured by the tachometer attached to the train and the GNSS receiver, it is judged to be an accurate value. Also, the accuracy of position measurement through on-site detection is within ±5 m.
Chapter 3.1.
Thank you for your comments for better research.
In this study, Wavelet Transform, not STFT, was applied as an analysis technique to find squats by analyzing acceleration data.
There are many cases and papers where Wavelet Transform is applied as a technique to evaluate the state of the orbit. A representative example is Professor Zilli's team at the Delft University of Technology in the Netherlands.
It would be appreciated if you refer to [25] among the references in this paper.
The test section of this study was a Korean general rail section (passenger/cargo mixed section), and the speed during the test was about 80 km/h and the vehicle was driven at a constant speed.
In order to focus on the contents of the squat, data from a section without a special track system such as a turnout was used.

Round 2
Reviewer 2 Report
The Author did not provide the manuscript with text in track changes mode. Also, the cover letter did not indicate what changes and where were made. Additionally, the Author Response File simply contains the text of the article, without any other comments.
The likely reason is that the changes to the content of the manuscript are practically none.
Still, a reader outside of Korea, unfamiliar with all models of railroad vehicles, is not able to learn much about the measurements. Are accelerometers standard equipment of the measuring vehicle? Most likely yes. So is it possible to repeat the experiment elsewhere?
I got answers to the questions asked (although not all), but they were not included in the text. The manuscript is very short and I see no contraindications for expanding the measuring part, which takes a dozen lines.
After expanding the description of the experiment, and not just processing the results, the article can be published as a research report. In my opinion, the scientific soundness is low.
Author Response
lines 93-94
The EM-140 is a vehicle that inspects Korea's track infrastructure.
lines 102-107
The GNSS system used in this study is the accuracy of 0.01m + 1ppm CEP as a high-precision GNSS RTK receiver industry. The maximum speed of the test vehicle is about 140 km / h and vehicle speed of the study period is about 80km / h. Speed measurement is determined as the correct value because it obtained the coordinate measured by the tachometer and the GNSS receiver attached to the train. In addition, the accuracy of the position measurement by the field inspection is less than ± 5 m.
lines 125-129
we applied the Wavelet Transform by analyzing the acceleration data in the present study not the STFT analysis techniques to find the squat. A technique for assessing the condition of the track and a lot of practice papers applying the Wavelet Transform. Representative examples include Professor Zili Li's team at Delft University of Technology in the Netherlands and Hitoshi in Japan. I would appreciate a note of the reference of the paper [25].
lines 187-190
The test section of this study was a Korean general rail section (passenger/cargo mixed section), and the speed during the test was about 80 km/h and the vehicle was driven at a constant speed. In order to focus on the contents of the squat, data from a section without a special track system such as a turnaround system was used. In order to focus on the contents of the squat, data from a section without a special track system such as a turnaround system was used.
Thank you for what you pointed out for better research.

Round 3
Reviewer 2 Report
The article can be published as a research report. In my opinion, the scientific soundness is low.